# Scalable Particle Generation
# for Granular Shape Study

**Yifeng Zhao**
Zhejiang University & Westlake University

**Jinxin Liu**
Westlake University

**Xiangbo Gao**
Westlake University

**Sergio Torres**
Westlake University

**Stan Z. Li**
Westlake University *

## Abstract

The shape of granular matter (particle) is crucial for understanding their properties and assembly behavior. Existing studies often rely on intuitive or machine-derived shape descriptors (*e.g.,* sphericity and Corey shape factors). Typically, these studies focus on single, individual particles with specific shape features, and statistical evaluations involving a large number of particles are rare due to the scarcity of particle samples. It raises doubts about whether the pre-selected shape descriptors can adequately capture the rich morphological information offered by particles. In this paper, we propose a two-step particle generation pipeline to evaluate the quality of the previous shape descriptors. First, we explicitly use a Metaball-Imaging algorithm to transform the pixel data into a lower-dimensional space. Secondly, we introduce a conditional generative method to design 3D realistic style particles. Meanwhile, we also design a new shape estimator to provide shape constraints to guide the conditional generation process. Building on this, we introduce the concept of "attribute twins" - particles that share identical shape features but differ in actual morphologies. They provide crucial particle samples to investigate whether existing shape descriptors are sufficient to represent the effects of particle shape. In a series of simulations focusing on the drag force experienced by settling particles in a fluid, we generate particle samples under different constraints of single or multiple shape descriptors. Our results shed light on the limitations of current shape descriptors in representing the influence of particle shape on this physical process and highlight the need for improved shape descriptors in future.

## 1 Introduction

Granular matter (particle) stands as one of the most abundant forms of matter globally, ranking second only to fluids in terms of their prevalence [1]. Commonly, they span a wide spectrum of particles and scales, encompassing entities ranging from viruses, cells to plastics and soil particles [2]. Across such a diverse spectrum, granular matter exhibits a multitude of complex physical and mechanical properties. It has been widely recognized that the macroscopic behavior of granular media is intricately linked to the micro-structural characteristics of the constituent particles. Among these characteristics, particle shape (morphology) emerges as one of the most influential factors in shaping the properties of granular matter. It governs not only granular traits (including friction, interactions, and deformations [3, 4]), but also profoundly influences the assembly responses (encompassing permeability, strength, and failure [5, 6]). Therefore, how to effectively characterize the particle shape is significant and fundamental for granular matter and understanding their behaviors.

---

*Corresponding author: `Stan.ZQ.Li@westlake.edu.cn`

NeurIPS 2023 AI for Science Workshop.

However, defining and characterizing the impact of shape is effort-intensive and places a significant burden on (expert) users to depict the particle shape impact for physical and chemical processes that they care about. This challenge is further exacerbated as granular matter often resides within a high-dimensional function space. In order to reduce dimensionality and simplify the granular matter modeling problem, existing methods [7] typically rely on intuitive or machine-picked shape descriptors (*e.g.,* roundness, convexity, and aspect ratio) and represent the unknown particle landscape with these learned parametric shape descriptors. However, it naturally raises the question: are these shape descriptors sufficient to capture the full extent of morphological impacts?

With the rapid advancement of numerical simulation methods in recent years [1, 8, 9], one may expect modern simulation methods to be beneficial for answering the above question and characterizing the influence of shape factors better. Unfortunately, their (numerical simulation) success often hinges on the availability of large amounts of data, while *the limited availability* of granular matter samples places a significant burden on users to characterize all degrees of variation, or may produce poor generalization along the axes that are not varied. More specifically, such limitation encompasses the scarcity of particle collections with continuous shape descriptors and leaves the presence of granules that share identical shape parameters but exhibit varying actual shapes. Thus, these constraints impede our capacity to obtain a more comprehensive understanding of the impact of particle shapes, leading to the degeneration of particle characterization.

In this paper, borrowing the idea of conditional generative models, we propose a particle generation method called Scalable PaRtIcle Generation (SPRIG) to address the above sample limitation. In particular, we propose a two-step particle generation pipeline. Our main contributions are as follows: 1) We first propose a dimension reduction method, namely Metaball-Imaging (MI), which can transform 3D granular pixel data into interpretable, low-dimensional sequences in the form of Metaball functions, while retaining a high degree of fidelity to the original particles. 2) On this basis, we develop a conditional generative adversarial network (cGAN) that can generate three-dimensional granular particles with specific geometric parameters by explicitly conditioning on certain shape descriptors, such as sphericity, circularity, Corey shape factors, or their combinations. 3) We demonstrate that the particles generated by SPRIG retain the authentic granular style observed in the input training data.

## 2 Related Work

### 2.1 Simulation Frameworks for Complex Shaped Particles

The intricate nature of granular matter presents significant challenges to analytical methods due to the associated complexities in particle shape. Consequently, numerical simulations have become an indispensable tool for unraveling the dynamics of particles with complex shapes [10–12], where the Discrete Element Method (DEM) is often used. DEM treats particles as rigid entities and considers their motions based on the Newton-Euler equations [13]. Many successful applications have been made to model realistic particles with DEM using X-ray computed tomography (XRCT), differing mainly in shape reconstruction methods and contact frameworks.

For example, Spherical-Harmonic (SH) DEM uses SH functions for shape and inter-particle contact detection [14]. Level-set DEM uses level-set functions and look-up mechanisms [15]. Signed Distance Field (SDF) DEM uses SDF functions and energy-saving contact theory [16]. Recently, the Metaball function has also been implemented to reconstruct realistic particle morphologies [17] and coupled with DEM in a gradient-based method [11]. This Metaball-based DEM approach strikes a balance between accurate particle shape representation and computational efficiency. Of particular note is its integration with the Lattice Boltzmann method, making it suitable for the study of fluid-particle systems with a significant number of complexly shaped particles [18].

### 2.2 Particle Generation Methods

Accurate reconstruction of granular matter requires the XRCT technique, which is time-consuming and costly. In practical engineering, only a small number of particles (less than 10% as reported in [19]) can be scanned. Direct simulation with them will suffer from repetitive particle morphologies. It is therefore necessary to generate realistic particles with co-essential morphological features.

Previous particle generation studies typically use a compressing-sampling approach [20, 21]. They compress particle representations (*e.g.,* XRCT images) into some control variables (*e.g.,* the SH

function) and add randomness to generate new particles. However, direct sampling of these variables can lead to underfitting or overfitting problems [22]. This is partly addressed by the use of mixture models [23, 21]. More recently, Variational Auto-encoders (VAE) have been used [22, 24], which provide a more flexible generative approach with better performance. More importantly, it allows a high level of control over the morphology of the generated particles [17], demonstrating the potential of deep learning techniques on this problem.

However, generating particles with co-essential morphological features is not sufficient. To gain a deeper understanding of the effect of particle shape and to assess the adequacy of existing shape descriptors in representing these effects, a method capable of generating diverse realistic-style particles with specific geometric features remains essential.

## 3 SPRIG: Scalable Particle Generation

### 3.1 Preliminaries

To understand the properties (granular traits and assembly responses) of granular matter[2], we can generally analyze them in three ways: 1) analytical method, 2) experimental method, and 3) simulation method. The analytical method is to use mathematical models to predict the behavior of granular matter, *e.g.,* using Newton's laws of motion. The experimental method solves this by conducting physical experiments in controlled laboratory environments. The numerical method focuses on computational models (*e.g.,* discrete element method) to explore the property and response of granular matter. In this paper, we will focus on the numerical simulation to characterize the particle.

However, the numerical simulation method requires a sufficient number of particle samples with different and specific geometric shape characteristics. Previous studies have attempted many particle reconstruction techniques that can introduce realistic particle shapes into the numerical framework. Such a technique often requires X-ray computed tomography (XRCT), which is time-consuming and costly. The large amount of resources required may not be accessible to all individuals and groups. It is therefore crucial to have a method that can learn from limited particle XRCT samples to generate a large number of new samples to meet research needs.

Then, we can identify a conditional generative task for numerical modeling. Formally, given a data set $\{\mathbf{x}\}$ of XRCT samples (voxeled data), we first build a labeled data set $\mathcal{D} := \{(\mathbf{x}, \mathbf{y})\}$, where $\mathbf{x}$ represents the normal XRCT samples and $\mathbf{y}$ is the computed shape features, *i.e.* $\mathbf{y}$ can represent sphericity, volume, surface area, and so on. Thus, we can define the conditional generative task as learning a conditional model $p(\mathbf{x}|\mathbf{y})$ to model the particle generation process.

### 3.2 Overall Pipeline

Typically, realistic particle data $\mathbf{x}$ is in voxel form, which has no inherent order or sequence and requires a lot of memory, making it impractical to tackle high-resolution scenes and difficult to learn. In addition, preprocessing of voxel data can be complex and computationally intensive. Tasks such as rotation, segmentation, and feature extraction often require specialized algorithms and significant computing resources.

Thus, to overcome the above challenges, we propose a two-step generative process. As shown in Figure 1, our pipeline consists of two steps, including the Metaball-Imaging and a conditional generation process. In the first step, the XRCT images of the particles $\mathbf{x}$ are transformed into (low-dimensional) Metaball avatars (See Appendix. A for details) $\bar{\mathbf{x}}$ using the Metaball-Imaging algorithm, thus obtaining $\bar{\mathcal{D}} := \{(\bar{\mathbf{x}}, \mathbf{y})\}$. In the second step, a conditional GAN is designed to solve this particle generation task in a low-dimensional space. Further, we also introduce a new shape estimator to improve the learning and enhance the interpretation of particle shapes.

### 3.3 Metaball-Imaging

The proposed Metaball-Imaging (MI) consists of three main parts: 1) data preprocessing, 2) acquisition of the principle outer contour with Sphere Clustering (SC), and 3) refinement of the distilled contour with Gradient Search (GS). We show the general framework in Figure 1 (*top*).

---

[2]In this paper, we use the terms "granular matter" and "particle" interchangeably.

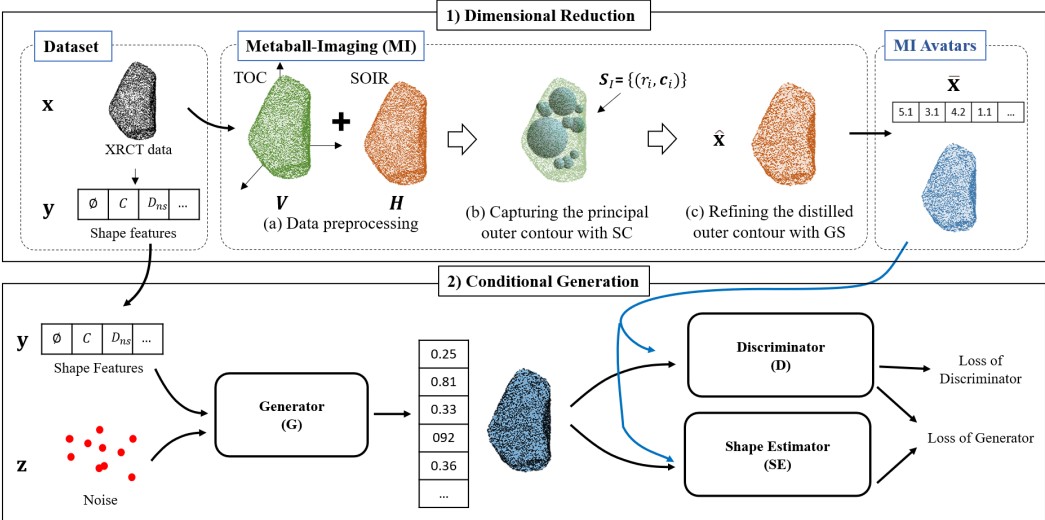

Figure 1: The overall framework of our SPRIG. It consists of two steps: First, we reduce the dimension of the input XRCT data into Metaball avatar using Metaball-Imaging algorithm. Then, an adaptation of a conditional generative adversarial network is implemented to perform the generation task.

**Data preprocessing**. We employ two preprocessing methods (Figure 1, *top (a)*): transformation of coordinates (TOC) and specification of interested region (SOIR). TOC is implemented to translate the XRCT voxels into the coordinate system centered at the origin. Such an operation is dedicated to obtain centralized voxelated representation $V$, which can avoid abnormal fitted parameters caused by XRCT coordinates. SOIR is aimed to distill the surface points $H$ from $V$.

**Capturing principle outer-contour with sphere-clustering (SC)**. The outer contour is captured by searching a series of non-overlapping inscribed spheres as control spheres (Figure 1, *top (b)*). This algorithm (Algorithm 1) starts with the Euclidean distance transformation [25] on the preprocessed voxelised particle $V$. It identifies the radius $r_i$ and center $c_i$ of the largest inscribed sphere by locating the maximum value in the transformed vector and its position. Voxels inside the inscribed sphere are then set to zero. This process is repeated until the number of inscribed spheres equals the number of control points $n$ (humanly selected), yielding a set of inscribed spheres denoted as $S_I$.

---

**Algorithm 1** The Sphere-Clustering (SC) Algorithm for the capture of the principal outer contour.

---

1: **Require:** the voxelized particle $V$, the number of control points $n$.
2: **for** $i = 1, 2, ...,$ to $n$ **do**
3:     *Transform* **-** Implementing Euclidean distance transform on $V$.
4:     *Search* **-** Finding the radius $r_i$ and center $c_i$ of maximum inscribed sphere with the maximum value in the transformed vector.
5:     *Reset* **-** Zeroing those voxels of the searched inscribed circles and updating $V$.
6: **end for**
7: **Return:** the distilled set of inscribed spheres $S_I = \{(r_i, c_i)\}, i = 1, \ldots, n$.

---

**Refining the distilled contour with gradient-search (GS)**. Then, we refine the outer contour using gradient search (Figure 1, *top (c)*). With the distilled inscribed spheres $S_I$, we create a Metaball model $\hat{x} = \{(r_i, c_i)\}$ to represent the primary outer contour of target particles. We then define a piecewise loss function, $L(\hat{x})$, to calculate gradient information based on the distilled point hull $H$ instead of the traditional Mean Square Error (MSE) to avoid distorted Metaball models and improve the adaptability to complex geometry:

$$L(\hat{x}) = \begin{cases} \sum_{i=1}^{n} (f_i^l(\hat{x}) - 1)^2, & f_i^l \in [2, +\infty) \\ \sum_{i=1}^{n} (f_i^l(\hat{x}) - 1), & f_i^l \in [1, 2] \\ \sum_{i=1}^{n} \left[ (f_i^l(\hat{x}) - 1)^2 + \frac{1}{f_i^l(\hat{x})} - 1 \right], & f_i^l \in [0, 1] \end{cases} \tag{1}$$

where $f_i^l(\hat{\mathbf{x}}) = \sum_{j=1}^m \frac{r_i}{(\boldsymbol{H_j} - \boldsymbol{c_i})^2}$ and $m$ is the number of surface points (we treat it as a constant).

Finally, we can optimize model parameters through gradient descent:

$$\bar{\mathbf{x}} \leftarrow \hat{\mathbf{x}} - \eta \cdot \nabla_{\hat{\mathbf{x}}} L(\hat{\mathbf{x}}), \tag{2}$$

where $\eta$ is the learning rate and $\nabla_{\hat{\mathbf{x}}} L(\hat{\mathbf{x}})$ is the gradient of $L(\hat{\mathbf{x}})$ with respect to the input $\hat{\mathbf{x}}$. We use a combination of Adam and SGD for the gradient updates to strike a balance between efficiency and convergence capability [26]. The process continues until the termination condition is met, which is determined by the number of generations $E^{gs}$ (see Algorithm 2).

---

**Algorithm 2** The Gradient Search for refinement of outer contour.

---

1: **Require:** the particle point hull $\boldsymbol{H}$, the number of generations $E^{gs}$, the learning rate $\eta$, the distilled set of inscribed spheres $\boldsymbol{S_I}$.
2: $\boldsymbol{S_I}$ is taken as the Metaball model of principle outer-contour, the initial value $\hat{\mathbf{x}}$.
3: **for** $i = 1, 2, ...,$ to $E^{gs}$ **do**
4: $\quad \bar{\mathbf{x}} \leftarrow \hat{\mathbf{x}} - \eta \cdot \nabla_{\hat{\mathbf{x}}} L(\hat{\mathbf{x}})$.
5: **end for**
6: **Return:** the searched parameter $\bar{\mathbf{x}}$.

---

## 3.4 Conditional Generation

Here we employ conditional generative networks (cGAN) to conduct the conditional particle generation [27, 28]. We take the distilled Metaball avatar $\bar{\mathbf{x}}$ and the corresponding shape features $\mathbf{y}$ as training data, *i.e.,* $\mathcal{D} := \{(\bar{\mathbf{x}}, \mathbf{y})\}$. As shown in Figure 1 *bottom*, we learn a generator and a discriminator, where the generator outputs a fake sample from a (vector) noise instance $\mathbf{z}$ and the conditional variable $\mathbf{y}$, and the discriminator distinguishes input samples as real or fake. Specifically, we train the discriminator with the following adversarial loss:

$$L_D = -\mathbb{E}_{\bar{\mathbf{x}}, \mathbf{y} \sim \bar{\mathcal{D}}}[\log D(\bar{\mathbf{x}}, \mathbf{y})] - \mathbb{E}_{\mathbf{z}, \mathbf{y}}[\log(1 - D(G(\mathbf{z}, \mathbf{y}), \mathbf{y}))]. \tag{3}$$

To train the generator, in addition to the standard discriminator-guided objective, we introduce two additional components ($L_{real}$ and $L_{shape}$):

$$L_G = w_1 * \mathbb{E}_{\mathbf{z}, \mathbf{y}}[\log(1 - D(G(\mathbf{z}, \mathbf{y}), \mathbf{y}))] + w_2 * L_{real} + w_3 * L_{shape}, \tag{4}$$

where $L_{real} = \frac{1}{N} \sum_{i=1}^N (G(\mathbf{z}, \mathbf{y}) - \bar{\mathbf{x}})^2$ accounts for producing real samples, which serves to stabilize the training process and prevent significant deviations between the generated output and the input data [29], and $L_{shape}$ is provided by a shape estimator (see below). The weights $w_1$, $w_2$, and $w_3$ are dynamically adjusted to balance the contributions of the various loss components, enhancing the overall performance of the model while maintaining stability.

**Shape estimator.** Since the generated result takes the form of a Metaball model corresponding to a specific 3D particle, its shape features can be inferred from the corresponding cloud points of that particle. However, this approach isn't compatible with the training of SPRIG, making it difficult to incorporate shape constraints into the optimization of the generator. Therefore, we introduce the Shape Estimator - a neural network based surrogate model - to predict accurate shape features from the generated results. It is seamlessly integrated into the computational graph of the generator network, providing morphological information for the generated results and guiding their training. The training of the shape estimator is based on the dataset $\mathcal{D} := \{(\bar{\mathbf{x}}, \mathbf{y})\}$. In particular, instead of direct training, we use a variational autoencoder (VAE) to distill regularised latent embeddings of $\bar{\mathbf{x}}$. We then make predictions based on $z$ using a Multilayer Perceptron (MLP).

## 4 Experiments

**Implementation details.** In MI, we set the Metaball parameters with $n = 40$, $\eta = 0.001$, and $E^{gs} = 5000$. The neural network models used for training are built using Pytorch. We apply MLP as the base model of the generator and discriminator. As for the shape estimator, we compare the VAE-MLP with various models and the quantitative comparison result can be found in Appendix B.

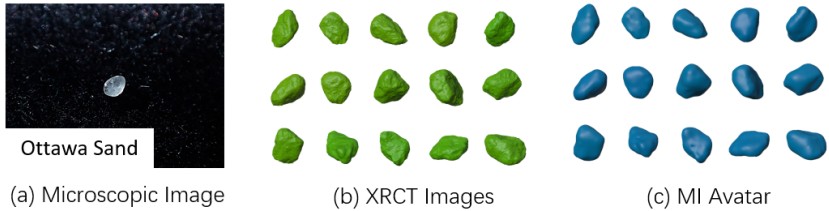

(a) Microscopic Image    (b) XRCT Images    (c) MI Avatar

Figure 2: Representative particles, XRCT images (green particles), and corresponding MI avatars (blue particles) of the Ottawa sand.

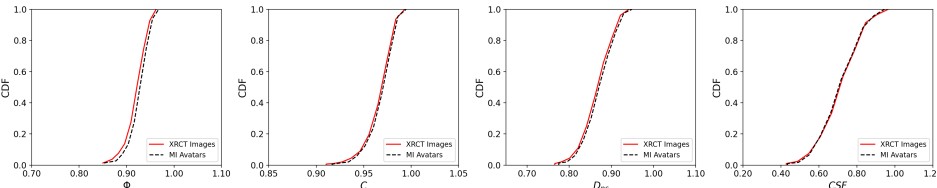

Figure 3: The Cumulative Distribution Function of the selected shape features of the Ottawa sand.

## 4.1 Datasets

As for the dataset, $\mathcal{D} := \{(\mathbf{x}, \mathbf{y})\}$, $\mathbf{x}$ stands for the XRCT image of the Ottawa sand and $\mathbf{y}$ stands for the selected shape features. In this paper, $\mathbf{x}$ consists of 290 Ottawa sands, which is a typical grain of pure quartz mined in Ottawa, Canada, as shown in Figure 2 (a). Due to geological transport, it has smooth and angled characteristics. For XRCT imaging, the ZEISS Xradia 610 Versa is used. The voltage of the X-ray source is set to 80 kV. The 0.4X detector is selected in the scan recipes, which means that the corresponding optical magnification is 0.4. The voxel size is 18.56 $\mu m$. On average, the particle in XRCT images contains more than $7.9 \times 10^6$ voxels to represent a real grain geometry. Particle segmentation is performed by "ilastik", a machine learning driven edge detection algorithm for XRCT images [30]. The imaged particles are shown in Figure 2 (b) with green particles. As for the shape features, we consider four dimensionless shape factors: sphericity $\phi$ and circularity $C$ [17], a combination $D_{ns}$ of nominal diameter $D_n$ and surface equivalent sphere diameter $D_s$, and the Corey Shape Factor (CSF). The definitions of these shape descriptors can be found in Appendix C. These shape features are calculated directly with the $\mathbf{x}$, as shown by the black dashed lines in Figure 3.

## 4.2 Results of Metaball-Imaging

We use 40 control points to transform the samples presented in Section 4.1 into the Metaball avatar $\bar{\mathbf{x}}$, as shown in Figure 2 ($c$) with blue particles (They are only for visualization and those avatars are in the form of Metaball function). For better generation performance, we carry out data augmentation on the distilled $\bar{\mathbf{x}}$, where slightly modified synthetic data is introduced. Here we employ particle rotation and parameter shuffling, where particle rotation is a popular strategy based on the rotation invariant property, and parameter shuffling is a random recombination of $\{k_i, \boldsymbol{x_i}\}$ in the Metaball avatar $\bar{\mathbf{x}}$. Employing parameter shuffling is because changing the order of the control spheres does not change the corresponding Metaball model. Such processing can effectively avoid the overfitting problem and improve convergence performance. During augmentation, each particle is rotated 5 times and the corresponding Metaball parameter is shuffled 50 times. The augmented datasets then contain 145,000 Ottawa sand samples. We calculate the shape characteristics of these Metaball avatars through the corresponding point clouds as shown by the red lines in Figure 3. And a good agreement between the real particles and the captured Metaball avatars can be observed.

## 4.3 Conditional Generation with Different Shape Features

In this section, we examine at the conditional generation of MI avatars under different configurations of shape features. Figure 4 ($a$) illustrates the resulting generated avatars. It can be seen that the

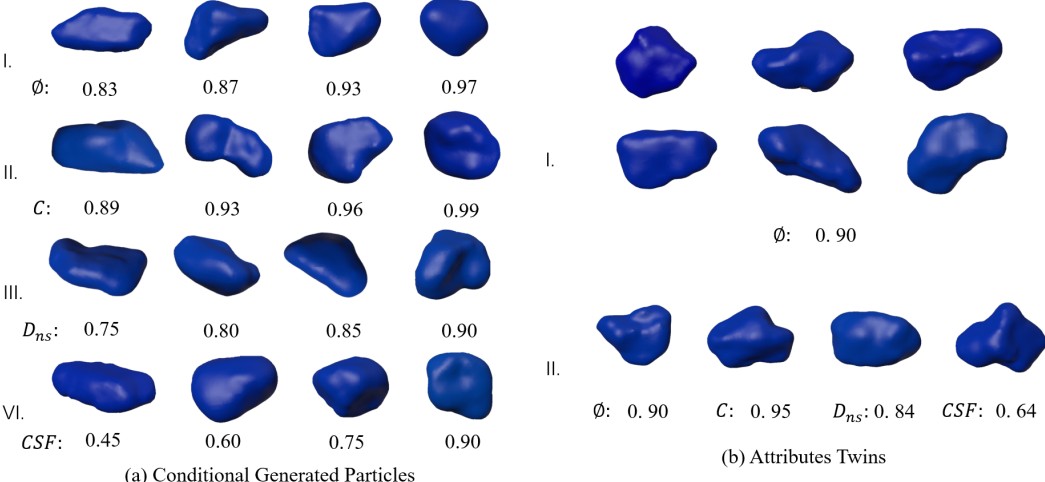

(a) Conditional Generated Particles

(b) Attributes Twins

Figure 4: The generated particles. (*a*) represent the conditionally generated particles of different shape features, where "I" denotes controls on $\phi$, "II" for $C$, "III" for $D_{ns}$, and "VI" for CSF. (*b*) stands for the attribute twins of different shape feature combinations, where all particles in "I" have the same $phi$ and particles in "II" have an identical set of four shape features.

generated particles maintain a coherent style inherited from their parent particles, showing continuous control over the selected shape features[3].

## 4.4 Attribute Twins

As shown in the previous section, SPRIG provides us with a tool for generating particles with precise shape features. This capability opens the door to generating particles with identical shape features but different morphologies, *i.e.,* attribute twins. Existing studies on the influence of particle shape focus on the incorporation of specific shape descriptors with a limited dataset of particle samples. However, it remains unclear whether these shape descriptors are sufficient to represent such an influence. And the attribute twins provide us with the opportunity to perform a comprehensive investigation of these shape descriptors.

Figure 4 (*b*) illustrates two sets of generated attribute twins that are in Metaball form. In the next section, we will select an important physical process to further investigate the above shape descriptors.

## 4.5 A Case Study on the Drag Force

The drag force is a fundamental resistance encountered by moving objects in a fluid. It exerts a significant influence on particle-laden flows, affecting phenomena like agglomeration, sedimentation, and diffusion. Understanding its interaction with particle shape remains a research challenge. Previous research has explored the integration of drag force coefficients with different shape descriptors. A popular choice is the sphericity [31–33]. To examine whether it can properly reflect the influence of particle shape, we carry out a series of simulations with the aforementioned attribute twins, employing the simulation framework previously developed by the authors[17] (we refer the reader to the definition and simulation setup in Appendix. C).

The result of the simulations for these two sets of attribute twins under different $Re$ conditions can be seen in Figure 5. It is interesting to note that although all the particles involved have the same sphericity, their drag force coefficient vary significantly across the different conditions. Even when we control for four specific shape descriptors (attribute twins shown in Figure 4 (*b*), II and result in Figure 5 II), the results obtained remain diverse. In particular, the scatter of results is more pronounced in high $Re$ conditions, as indicated by a higher relative standard deviation (RSD),

---

[3]We also point out that this method may suffer from cases of generation failure and precision errors, where the desired output may not be achieved. However, this can be mitigated by multiple training and generation in actual application (here we leave further analysis in the future work).

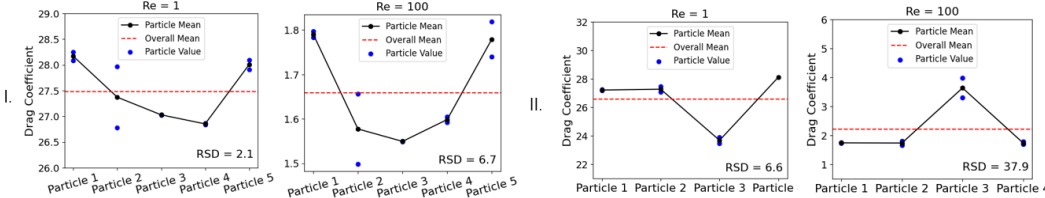

Figure 5: The simulation result of the generated two sets of attribute twins under different $Re$. "I" stands for the result of attribute twins shown in Figure 4 (*b*) I and "II" corresponds to Figure 4 (*b*) II.

compared to low $Re$ scenarios. Furthermore, the spread of results does not necessarily decrease as the number of controlled shape descriptors increases. This may be related to the fact that even when controlling for multiple shape descriptors, particles may still exhibit significant variation in their shapes. In conclusion, the above results suggest that sphericity, even when combined with several other shape descriptors, is not sufficient to provide a comprehensive account of the effect of particle shape on drag, indicating the need to develop a better shape descriptor or consider additional factors to gain a better understanding of the effect of particle shape.

## 5    Conclusion and Future Work

In this paper, we present SPRIG, a scalable particle generation method. SPRIG has the ability to learn from XRCT images, which are in voxel format, and subsequently generate realistic 3D particles with specific shape features. Building on this model, we introduce the concept of "attribute twins" - particles that share identical shape features but have different actual morphologies.

To investigate whether shape descriptors such as sphericity are sufficient to adequately represent the influence of shape in physical processes, we performed a series of simulations using these attribute twins on the drag problem, a crucial aspect of fluid-particle systems. Our results indicate that sphericity does not fully capture the influence of shape, even when controlling for multiple shape features with it. This discrepancy may be due to the limitations of these shape descriptors, which may not fully govern particle morphologies. Consequently, particles with identical sets of shape features may still exhibit variations in their actual shapes.

In the future, we plan to use the generated attribute twins in various physical problems to evaluate the effectiveness of classical attribute twins. Ultimately, our goal is to address the fundamental question: What really defines "shape" in these physical processes?

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

Table 1: Comparison of model performance on a held-out region

| Model | Train MSE | Test MSE | Train $R^2$ | Test $R^2$ |
|---|---|---|---|---|
| MLP | 0.0102 | 0.1621 | 0.90 | 0.88 |
| CNN | 0.0183 | 0.1154 | 0.91 | 0.85 |
| Transformer | 0.0089 | 0.1780 | 0.93 | 0.80 |
| VAE-MLP (Ours) | 0.0088 | 0.0091 | 0.97 | 0.95 |

## Appendix A. The Metaball avatar and Metaball function

The Metaball avatar $\bar{\mathbf{x}}$ is the parameter set of the Metaball function is a mathematical function which can represent complex shapes:

$$f(\boldsymbol{x}) = \sum_{i=1}^{n} \frac{\hat{k}_i}{(\boldsymbol{x} - \hat{\boldsymbol{x}}_i)^2} = 1 \tag{5}$$

where $\boldsymbol{x}$ is the 3D coordinate. $n$ denotes the number of control points. $\hat{\boldsymbol{x}}_i$ refers to the position of the $i$-th control point, which serves as a skeleton for the parameterized shape, and $\hat{k}_i$ represents the positive coefficient determining the weight of the $i$-th control point. This implicit, interpretable function defines shapes by a series of isosurfaces constructed from points in space that satisfy certain function values around particular control points (see Fig. 6). Points with function values of 1 are considered to be points on the surface of the particle, while those with values greater than 1 are inside the particle, and those with values less than 1 are outside the particle.

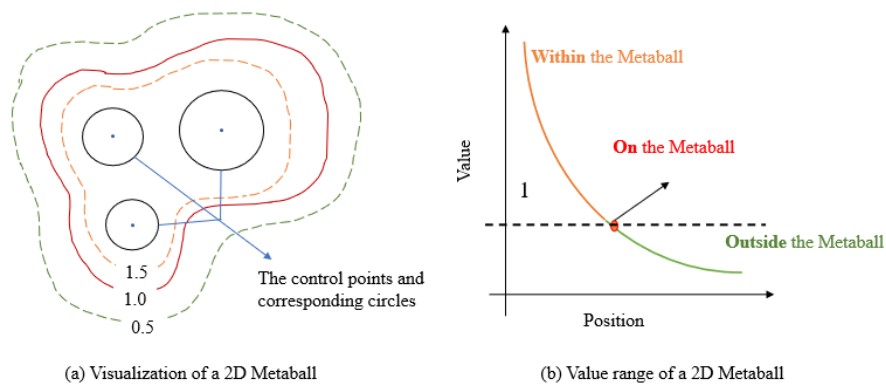

(a) Visualization of a 2D Metaball     (b) Value range of a 2D Metaball

Figure 6: The Metaball function.

## Appendix B. The quantitative comparison of different models for the shape estimator

Table 1 illustrate the comparison result of different model for the shape estimator. In the end, we select VAE-MLP as our preferred shape estimator.

## Appendix C. The definitions of utilized shape descriptors

The Corey Shape Factor $CSF$ reveals the dimension feature of the studied particle, as given by:

$$\text{CSF} = \frac{L_s}{\sqrt{L_i L_l}} \tag{6}$$

where $L_s$, $L_i$ and $L_l$ are the shortest, intermediate and longest axis lengths of particles.

The nominal diameter $D_n$ and surface-equivalent-sphere diameter $D_s$ are two widely used parameters. The $D_n$ is defined as the diameter of the volume-equivalent sphere. And the $D_s$ takes the following

form:

$$D_s = \sqrt{\frac{4A_p}{\pi}} \tag{7}$$

where $A_p$ = the maximum projected area of the particle. Here, they are combined as $D_{ns} = D_n/D_s$ to form a dimensionless quantity.

The sphericity $\phi$ is the measure of similarity between the studied particle and the sphere, which is defined as:

$$\phi = \frac{A_{ve}}{A} \tag{8}$$

where $A_{ve}$ = the surface area of the volume-equivalent sphere to the studied particle; $A$ = the surface area of the studied particle.

Another frequently used metric is the circularity $C$, which evaluates the roundness of non-spherical particle:

$$C = \frac{\pi D_s}{P_p} \tag{9}$$

where $P_p$ is the the perimeter of the particle's projected-area.

## Appendix D. The simulation setup for the examination of drag force

The simulation setup is shown in Figure 7 ($a$). The domain is set to have a constant inflow velocity $U_{in}(0.05\ m/s)$ and a gradient-free outlet with size: $0.30 \times 0.30 \times 0.30 m$. Periodic boundary conditions are applied to the rest. The density of the fluid is $1000\ kg/m^3$. The perturbation of the flow is controlled with $Re$ by varying the viscosity. Two states of Reynold number[4] $Re$ = 1 and 100 are implemented in the simulation. The volume of the particles is chosen to be equal to the sphere with radius $0.005m$ to avoid the domain size effect on the result and to fully reflect the effect of particle shape [34]. The LBM and DEM time steps are both set to $2.0 \times 10^{-4}s$. The LBM space step is set to $1.0 \times 10^{-3}s$. Here $Re$ is redefined as

$$Re = \frac{\rho_f U_f D_e}{\mu} \tag{10}$$

To include the effect of particle orientation with respect to the flow direction, these particles are placed with their maximum projected area perpendicular to the flow direction. For evaluation, the drag coefficients are calculated and compared. The drag force is calculated with the simulated hydrodynamic force $F_D$ along the flow direction:

$$C_D = \frac{F_D}{\frac{\rho_f}{2}U_{in}^2 A_{eq}} \tag{11}$$

where $A_{eq}$ is the cross-sectional area of the volume equivalent sphere whose radius is $0.005m$. Figure 7 (b) shows the typical result of the flow fields around the selected realistic particle.

---

[4]The Reynold number is a dimensionless parameter used in fluid dynamics to characterize the flow of fluids. It plays a crucial role in understanding the behavior of fluids in different situations, especially the drag force experienced by some object moving through a fluid.

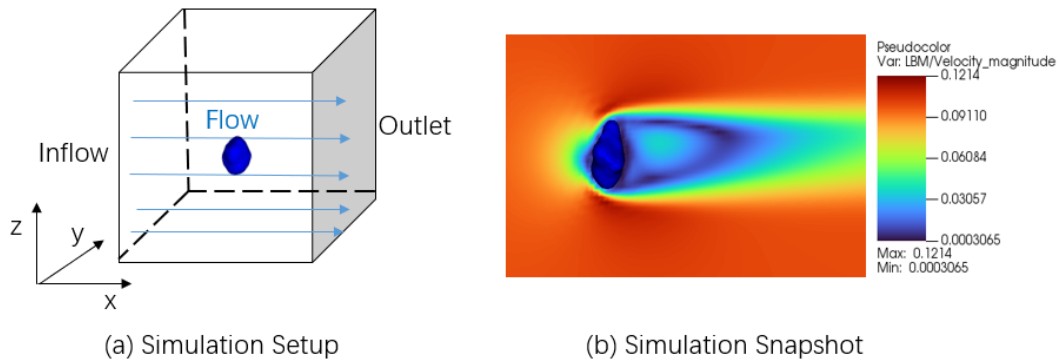

Figure 7: Configuration and snapshot of the simulation for assessing the drag force. The color in the background stands for the velocity magnitude of the fluid.

