# OpenReview forum: "Scalable Particle Generation for Granular Shape Study"
_NeurIPS.cc/2023/Workshop/AI4Science — NeurIPS2023-AI4Science Poster_

### Official Review · Reviewer_NJVR · 2023-10-11
**Review of Scalable Particle Generation for Granular Shape Study**

**Rating:** 7
**Confidence:** 2

**Review:**

Summary:
This paper presents the development of a Metaball-Imaging algorithm to transform pixel data into a lower-dimensional space vector followed by conditional generation using the descriptors as training examples to generate 3D particles. They also explore the generation of particles with the same shape features but different morphology and demonstrate limitations of using shape descriptors to predict physical processes.

Clarity and Quality:
This paper was clearly written with good quality figures. The details of the algorithm were also clearly described. They also effectively communicated the objective of their model.

Strengths:
Well written and well communicated. They successfully generate particles from pixel data with desired shape constraints that are similar to the XRCT images. They also show applications in their model for exploring the limitations of shape descriptors for predicting drag of a particle.

Weaknesses:
Lacks comparison to how their model generates from XRCT images vs. results from VAEs and compressed particle representations.

---

### Official Review · Reviewer_zEUN · 2023-10-22
**It seems a nice paper and I am positive to its acceptance. But the content is beyond my expertise.**

**Rating:** 7
**Confidence:** 2

**Review:**

The paper uses an interesting two-step generative pipeline to evaluate the generalizability of the shape descriptors. I appreciate the rigorous content put forth by the authors. I believe this submission aligns well with the quality of work expected for the AI4Science workshop.

However, it's worth noting that the scientific content within this paper exceeds the bounds of my domain expertise. Therefore, while I recognize its merit, my feedback is limited in scope and may not delve into the depth and intricacies of the topic.

I only have one question on the feature selection:

The paper underscores the application of the proposed pipeline for evaluating the efficacy of selected shape descriptors in capturing the necessary information for specific tasks. This naturally raises a question: how do the authors quantitatively assess the adequacy of these shape descriptors for the tasks in question? Providing a defined set of criteria or metrics for this evaluation would enhance the clarity and applicability of the study. A detailed discussion on this aspect would be a valuable addition.

---

### Meta-Review · Area_Chair_FG3p · 2023-10-27

**Recommendation:** Accept (Oral)
**Confidence:** 5

**Metareview:**

## Summary

In this paper, authors have developed a scalable particle generation method by utilizing low-dimensional representations from Metaball Imaging and applying a conditional generative network. The approach developed by authors leads a very good example of high-fidelity generative applications of latent space as shown for particle shape descriptors here but would be quite relevant to AI4Science community at large.
The paper is quite well written and is recommended for Oral presentation.


## Strength:
High-fidelity conditional generation of particles from pixel data with desired shape constraints.
Domain specific application of this model is also documented clearly as a case study.

## Weakness:
Comparison with other baseline models would help contextualize these results better.